# Construction of a Ginseng Root-Meristem Sensor and a Sensing Kinetics Study on the Main Nitrogen Nutrients

**DOI:** 10.3390/s21030681

**Published:** 2021-01-20

**Authors:** Shiang Wang, Dingqiang Lu, Guangchang Pang

**Affiliations:** Tianjin Key Laboratory of Food Biotechnology, College of Biotechnology & Food Science, Tianjin University of Commerce, Tianjin 300314, China; ronabell1219@163.com

**Keywords:** ginseng, nitrogen nutrients, continuous cropping, root meristem, electrochemical sensor

## Abstract

Severe continuous cropping obstacles exist in ginseng cultivation. In order to assess these obstacles, a “sandwich” ginseng root tissue sensor was developed for the kinetic determination of five nitrogen nutrients. The results showed that the sensing parameters of the sensor reached an ultrasensitive level (limit of detection up to 5.451 × 10^−24^ mol/L) for the five nitrogen nutrients, and exhibited good stability and reproducibility. In the order of two-, four-, and six-year-old ginseng plants, the sensitivity to inorganic nitrogen nutrients (sodium nitrate and urea) showed an upward trend following an initial decline (the interconnected allosteric constant Ka values acted as the parameter). The fluctuations in sensor sensitivity to organic nitrogen nutrients, specifically nucleotides (disodium inosinate and disodium guanylate), were relatively small. The sensor sensitivity of two-, four-, and six-year-old ginseng plants to sodium glutamate was 9.277 × 10^−19^ mol/L, 6.980 × 10^−21^ mol/L, and 5.451 × 10^−24^ mol/L, respectively. Based on the survival rate of the seedlings and mortality rate of the ginseng in each age group, a Hardy–Weinberg equilibrium analysis was carried out. The results showed that the sensing ability of the root system to sodium glutamate may be an important factor affecting its survival under continuous cropping obstacles with increasing age.

## 1. Introduction

Plants sense, absorb, and transport various nitrogen nutrients (amino acids, ammonium, nitrates, nucleotides, etc.) through the roots and synthesize carbohydrates and carbon skeletons through photosynthesis, thereby converting the nitrogen nutrients absorbed by the roots into amino acids and nucleotides—the basic components required for the synthesis of biological macromolecules [1]. Since the efficiency of carbon dioxide fixation in the air by plants through photosynthesis mainly depends on their own genetic basis, it is difficult to achieve artificial optimization and cultivation, and thus the yield of plants, especially crops, mainly depends on the research and improvements in cultivation techniques, such as irrigation, fertilization, soil improvement, and intercropping. However, no studies have extensively assessed how plant roots sense, absorb, and transport these basic nutrients and how plants achieve coordination and balance between inorganic nitrogen sources and carbon sources according to photosynthesis and their physiological and biochemical needs. One reason for this is that quantitative nutrient sensing and measurement technology, especially receptor-sensing technology, is lagging behind the human health and medical fields. Our research group has focused on biosensors [2,3] and tissue sensors, especially receptor sensors [4,5,6,7], and satisfactory results have been achieved in the construction of G protein-coupled receptor (GPCR) nano and tissue sensors, receptor gene expression, self-assembly, cross-species self-assembly sensors, and their quantitative and kinetic research. Using ginseng (*Panax ginseng* C.A.Mey) as the study material, we aimed to address the technical limitations in the quantitative determination of nitrogen nutrients using plant roots. An electrochemical-type biosensor was prepared by fixing the root meristems of two-, four-, and six-year-old ginseng plants, and the sensing and kinetic parameters of five important nitrogen nutrients, including sodium nitrate, urea, sodium glutamate, disodium guanylate, and disodium inosinate, were quantitatively determined through the electrochemical workstation and the ginseng root-meristem sensor. Our aim was to elucidate the sensing pattern and sensing mechanisms of ginseng root meristems of different ages to the five nitrogen nutrients and their association with continuous cropping obstacles.

Nitrate is regarded as one of the most important nitrogen sources for plants. It serves not only as a type of nitrogen source nutrient, but also as an important signal affecting plant nitrogen and carbon metabolism, as well as organ growth and development [8,9,10]. Urea is currently the most commonly used and most common inorganic nitrogen fertilizer that plays an important role in transamination, deamination, ornithine cycle, and arginine and nucleotide synthesis in higher organisms, especially higher animals. Thus, the present study first selected sodium nitrate and urea as sensing ligands and focused on the sensing patterns of ginseng root meristems. However, in order to provide the basic raw materials for the synthesis and replication of biological macromolecules (such as protein, DNA, and RNA) during plant growth, development, differentiation, and fruit development, the sensing and absorption of nitrogen sources by the roots need to be coordinated with the synthesis of the carbon skeleton of amino acids and nucleotides (photosynthesis and the Calvin cycle), and the metabolic balance between them needs to be maintained. Therefore, inorganic nitrogen sources can become true nitrogen nutrients only when they are compatible with the needs of plants for nitrogen sources; otherwise, they can only have toxic effects on plant growth (such as the excessive application of inorganic nitrogen fertilizers). As a result, we also selected sodium glutamate, the most important amino acid in amino acid metabolism, as the key conversion station for transamination reactions, which is an important ligand for the umami taste receptor T1R1. Concurrently, as the ligands for the sensing measurements and comparative analysis, we selected disodium inosinate, the most important compound in nucleotide metabolism, and disodium guanylate, which provides energy for protein synthesis, for the activation of signal transduction by G proteins (including: G protein heterotrimer Gαβγ and small G protein), and for transport into and out of the nucleus by decomposing guanosine triphosphate (GTP) into guanosine diphosphate (GDP).

## 2. Materials and Methods

### 2.1. Materials and Reagents

The ginseng plants used in the experiment all demonstrated good growth and were the same ginseng species produced from a plantation in Changbai Mountain, Jilin Province. Multiple two-year-old, four-year-old, and six-year-old ginseng seedlings were selected, and the lateral root meristems were used as the samples. Soluble starch was provided from Yingda Sparseness & Nobel Reagent Chemical Factory (Tianjin, China); glutaraldehyde was provided by Tianjin Bodi Chemical; sodium alginate was provided by Guangfu Fine Chemical Research Institute (Tianjin, China); CaCl_2_ was provided by Sigma Corporation (USA); the Nuclepore membrane was provided by Whatman (UK); disodium inosinate, disodium guanylate, sodium glutamate, and urea were provided by Sangong Biotech (Shanghai, China); NaNO_3_, HNO_3_, and absolute ethanol were provided by Guangfu Reagent (Tianjin, China); and phosphate-buffered saline (PBS) was provided by Tianshun (Shijiazhuang, China). All reagents were of analytical grade, and the water used in the experiments was ultrapure.

### 2.2. Instruments and Equipment

The instruments used in the experiment were as follows: analytical balance (Precision Scientific Instrument, Shanghai, China); Milli-Q Reference ultrapure water system (Yarong Biochemical Instrument, Shanghai, China); water bath (Yiheng Technology, Shanghai, China); magnetic stirrer (Honour Instrument, Tianjin, China); KQ 3200B ultrasonic cleaner (Kunshan Ultrasonic Instruments, Kunshan, China); CHl600E electrochemical workstation, three electrode system: glassy carbon electrode (φ = 3 mm), reference electrode: Ag/AgCl electrode, and platinum wire electrode (CH Instrument, Shanghai, China).

### 2.3. Pre-Treatment of Glassy Carbon Electrodes

After being polished with aluminum powder (α-Al_2_O_3_) on a chamois leather, the six glassy carbon electrodes were rinsed with ultrapure water and then cleaned in an ultrasonic water bath for 30 s. The procedure was repeated three times, following which the treated electrode was washed using 50% HNO_3_, absolute ethanol, and ultrapure water, sequentially.

After cleaning, a three-electrode system (the Ag/AgCl electrode as the reference electrode and the platinum wire electrode as the counter electrode) was used according to the existing method [11], and the glassy carbon electrode was activated by H_2_SO_4_ using cyclic voltammetry (scanning rate of 100 mV/s).

After the electrode was treated, one electrode was placed in a 1 × 10^−3^ mol/L K_3_Fe(CN)_6_ solution containing 0.20 mol/L KNO_3_ and analyzed by cyclic voltammetry (scanning rate of 50 mV/s). The pre-treatment effect of the glassy carbon electrode was characterized at −0.1 V to +0.6 V. The peak potential difference of the cyclic voltammetry curve of the glassy carbon electrode after pre-treatment was below 80 mV and was as close as possible to 64 mV when the electrode could be used. The other five pretreated glassy carbon electrodes were placed in a nitrogen environment to be dried for later use.

### 2.4. Preparation of a Ginseng Root-Meristem Sensor

Reagent preparation: The soluble starch was dissolved in an aqueous solution containing 1% glutaraldehyde, and the mixture was heated in a water bath at 80 °C and stirred for 30 min to prepare a starch solution at a concentration of 1%. The solution was allowed to stand overnight at room temperature, so that the starch and glutaraldehyde could fully crosslink to obtain the aldehyde-based starch gel solution. A 2% sodium alginate solution was prepared and stirred overnight with a magnetic stirrer, and a CaCl_2_ solution with a mass concentration of 5% was prepared.

Preparation of the ginseng root meristem: Three to five ginseng plants of the same age were obtained and cut into multiple lateral root tip meristems (length from 0.3 cm to 0.5 cm) with root hairs, and the samples were washed with ultrapure water, mixed, and cut into small pieces for later use.

Preparation of electrochemical biosensors: The aldehyde-based starch gel solution was mixed with the sodium alginate solution at a volume ratio of 1:1 [12,13,14] for future use. The prepared ginseng root-tip pieces were placed at the center of a Nuclepore membrane to form a dot of about 0.2 cm^2^. On the tissue dot, about 4 μL of the mixed solution was added, followed by the addition of about 4 μL of CaCl_2_ on the same tissue dot. This facilitated the sodium alginate and CaCl_2_ to undergo an ion exchange reaction to form a stable chelate, allowing the sodium alginate solution to gel into a good fixative agent [15,16]. Subsequently, the sample was covered with another sheet of Nuclepore membrane to make a “sandwich” structure.

Preparation before measurement: The measuring membrane with a sandwich structure was rinsed with ultrapure water to wash away the Cl^−^ and Ca^2+^ remaining on the membrane. Then, the ginseng root meristem at the center of the measuring membrane was aligned with the glassy carbon electrode, allowing the ginseng root meristem to overlap with the characterized electrode. Subsequently, the system was immobilized with a leather sheath to obtain the biosensor.

### 2.5. Measurement of the Ginseng Root-Meristem Sensor

A three-electrode system was adopted. The glassy carbon electrode with the immobilized ginseng root-meristem measuring membrane was used as the working electrode, the Ag/AgCl electrode was used as the reference electrode, the platinum wire electrode was used as the counter electrode, and ultrapure water was used as a blank control. Under a certain voltage (−0.38 V), the response currents of five solutions—disodium inosinate, disodium guanylate, sodium glutamate, sodium nitrate, and urea—were measured using a current–time measurement method, the change rate of the response current was used as the detection index, and measurement was carried out three times in parallel at each concentration. The equation for calculating the change rate of the response current was as follows:ΔI = (I_1_− I_2_)/I_1_,(1)
where ΔI represents the change in the response current, I_1_ represents the response current value (A) of the blank control, and I_2_ represents the response current value (A) of the ligand solution.

After being polished with aluminum powder (α-Al_2_O_3_) on a chamois leather, the glassy carbon electrode was rinsed with ultrapure water and then cleaned in an ultrasonic water bath for 30 s. The procedure was repeated three times, following which the treated electrode was washed using 50% HNO_3_, absolute ethanol, and ultrapure water, sequentially.

Figure 1 shows the diagram of the measurement principle of the entire assembly of the ginseng root-meristem sensor. After the ginseng root meristem was assembled on the electrode as a sensitive element, the five important nitrogen-source ligand solutions were detected. During the detection process, the extracellular domain of the plant-sensing receptor and the nitrogen source ligand allosterically interact with each other, causing the intracellular domain to change and alter the ion channel to transmit signals to the cells. The electrochemical workstation captures this non-genomic “fast pathway” signal, and the electrical signal released by the allosteric linkage between the sensing receptor and the ligand is transduced to the computer for signal recording. The sensing sensitivity of the ginseng meristem to different nitrogen source ligands can be preliminarily and quantitatively detected, and the pattern of interaction between the sensing receptor and nitrogen source ligands can be explored.

## 3. Results and Discussion

The core of biological tissues, cells, or receptor-sensing measurements is the selection of biologically sensitive materials (or elements). According to the functions and characteristics of plant roots, this study used the assembly and detection methods of animal tissue sensors as a reference and selected lateral root tip meristems as the sensitive materials. Through electrode pre-treatment, the root meristem was immobilized between two Nuclepore membranes, following which the system was sealed on the pretreated and characterized electrode, and measurements were carried out by an electrochemical workstation. In order to ensure that the measured electrochemical signal was the signal output by the root meristem under the recognition and stimulation of ligands (different nitrogen sources) and avoid the interference of various noises, we characterized every step of the assembly.

### 3.1. Electrode Pre-Treatment and Its Characterization

In the electrochemical analysis of animal taste bud sensors, cyclic voltammetry has laid the foundation for the sensing experiment method [2,17]. The glassy carbon electrode (GCE) is activated by H_2_SO_4_ to generate negatively charged oxygen-containing groups (such as hydroxyl and carboxyl) on the surface [11], thereby forming a porous structure and increasing the effective surface area [18].

Before and after this pre-treatment method, the GCE was characterized by cyclic voltammetry (Appendix A) and scans were made with the cyclic voltammogram at different rates Appendix A).

### 3.2. Assembly and Characterization of Sensitive Elements of the Ginseng Root Meristem

Figure 2 shows the characterization results of the bare electrode, bare electrode + Nuclepore membrane, and bare electrode + Nuclepore membrane + ginseng root meristem in 1mmol/L K_3_Fe(CN)_6_ (containing 0.20 mol/L KNO_3_). The comparison of the three curves in the figure revealed that the currents with successively assembled Nuclepore membranes and the ginseng root meristems exhibited an apparent and gradual decline. After the ginseng root-meristem membrane was assembled on the GCE, its electron transfer was obstructed, indicating that the ginseng root-meristem measuring membrane constitutes a biologically sensitive detection layer (element) that is able to avoid the interference caused by the direct contact between the analyte and electrode. That is, the biologically sensitive element was successfully assembled on the electrode.

### 3.3. Potential Optimization of the Current–Time Measurement Method

The current–time method was used to measure the ginseng root-meristem sensor at different potentials (Appendix A).

### 3.4. Determination of Five Types of Nitrogen Source Sensing Parameters by Ginseng Root-Meristem Sensors of Different Ages

Recent botanical research has focused on how plants can sense, transport, and absorb nutrients needed from the surrounding environment, as the sensing and absorption of nitrogen source nutrients by plant roots is an important guarantee for supplying the nitrogen sources required for normal plant growth. Research on the utilization of nitrogen sources and the supporting effect of nitrogen sources on plant growth has accumulated rapidly. For example, the plant Arabidopsis thaliana has at least two receptors (NRT1.1 [19] and NRT2.1 [20,21]) involving nitrates. P2K1 is an essential receptor for extracellular purines (ATP) in A. thaliana [8,22], and glutamate receptors (GLRs) in A. thaliana are related to defense signals [23,24,25]. Some plants absorb amino acid molecules through glutamic oxaloacetic transaminase (GOT) [26], and it has been reported that the ginseng amino acid transporter gene is PgLHT [27]. Most of these studies have focused on the transport of nitrogen sources and signal transduction in plants, while there are comparatively fewer reports on the sensor receptors of nitrogen sources and their quantification methods.

Taking into account the previous report experience [17] and the principle of interaction between receptor and ligand, it is believed that when plant sensing receptors recognize nitrogen source ligands, they do not recognize the N element in the molecule/ion, but instead recognize the entire molecule/ion structure. Therefore, the target measurements should be calculated using the concentration of molecules/ions. In this study, prepared ginseng root-meristem sensors of different ages were placed in five solutions (sodium nitrate, urea, sodium glutamate, disodium inosinate, and disodium guanylate) at different concentrations. Measurement was carried out in an order of 1 × 10^−20^ mol/L, 1 × 10^−19^ mol/L,…, 1 × 10^−10^ mol/L to determine the limit of detection. The time–current method was adopted, and the scanning potential was set to −0.38 V. After the system was allowed to stand for 10 s, the steady-state current value (120 s) was used as the comparison standard, and a graph was plotted with the change rate of the response current ΔI in sensing the nitrogen source on the *y*-axis (Appendix A).

The sensing of sodium nitrate by ginseng plants of different ages was used as an example. The minimum detection concentrations of the root meristem sensors of two-, four-, and six-year-old ginseng plants to the sodium nitrate solution were 1 × 10^−20^ mol/L, 1 × 10^−23^ mol/L, and 1 × 10^−20^ mol/L, respectively (Appendix A).

Figure 3A shows the correlation between the response current of the root meristem sensors of two-, four-, and six-year-old ginseng plants and the concentration of sodium nitrate. By fitting this curve, it was found that there was a good hyperbolic relationship between the change rate of response current and the sodium nitrate concentration. Furthermore, the sensors of the ginseng plants of different ages exhibited the same trend and characteristics with the response current of the sensor in detecting the concentrations of the other four nitrogen nutrients (urea, sodium glutamate, disodium inosinate, and disodium guanylate) (Appendix A).

Figure 3B shows the linear relationship between the change rate of the response current (ΔI) and the sodium nitrate concentration. According to a large amount of work published in the past [2,4,5,6,7], the double-reciprocal method was used to process the data for linear fitting, and the linear equations were as follows:

two-year-old ginseng:(2)$$\frac{1}{\Delta I}=1.24921+1.9254\times 10^{-21}\frac{1}{C}\;(R^{2}=\;0.99263)$$
four-year-old ginseng:(3)$$\frac{1}{\Delta I}=1.05223+1.0744\times 10^{-24}\frac{1}{C}\;(R^{2}=\;0.96975)$$
six-year-old ginseng:(4)$$\frac{1}{\Delta I}=1.13975+9.882\times 10^{-22}\frac{1}{C}\;(R^{2}=\;0.98209)$$

According to this equation, the interconnected allosteric constant Ka values of the interaction between sodium nitrate and the root meristem sensors of the two-, four-, and six-year-old ginseng plants could be calculated, and they were 1.491 × 10^−20^ mol/L, 6.529 × 10^−24^ mol/L, and 8.315 × 10^−21^ mol/L, respectively.

Furthermore, the sensing kinetic parameters of the ginseng root meristems of different ages to the other four nitrogen nutrients (urea, sodium glutamate, disodium inosinate, and disodium guanylate) were measured (Appendix A). After linear fitting, all R2 were greater than 87%. The obtained interconnected allosteric constant Ka values are shown in Table 1. These results indicated that the ginseng root-meristem sensor provides a detection method to understand the sensing ability of the root system. Taking sodium nitrate as an example, compared with other methods for detecting sodium nitrate, this sensor has higher sensitivity, and the detected plants are more targeted, as shown in Table 2.

Figure 4 shows the comparison of the sensing sensitivities (1/Ka) of the root meristems of the two-, four-, and six-year-old ginseng plants to five nitrogen sources, and different bars represent five different nitrogen sources. Since a smaller interconnected allosteric constant Ka value means that the corresponding sensitivity is higher, the reciprocal of this graph was created.

Meanwhile, in the horizontal comparison of five kinds of nitrogen nutrition, it can be seen intuitively that disodium inosinate and disodium guanylate had the same sensing trend, and the sensitivity fluctuation was not distinct. The sensing trends of sodium nitrate and urea were the same, and the sensing sensitivity of the four-year-old ginseng was the highest; only the sensing sensitivity of sodium glutamate increased gradually with increasing ginseng age.Different nitrogen nutrients are many orders of magnitude different from each other. The above results all show that the ginseng root-meristem sensor has good selectivity.

### 3.5. Reproducibility and Stability of Ginseng Root-Meristem Sensors

We put the measured ginseng root-meristem sensor vertically into a 1.5-mL centrifuge tube with ultrapure water and soaked it at 4 °C for 4–5 h. After soaking, we rinsed the sensor with ultrapure water. The flushed sensor continued to measure the next detection target. This shows that the ginseng tissue sensor has good reproducibility.

The ginseng root-meristem sensor was continuously measured 10 times in the same concentration of sodium nitrate solution, and the current change rate was 4.73%, indicating that the sensor was stable. The same constructed sensor was stored in ultrapure water at 4 °C, and the same concentration of sodium nitrate solution was measured every 24 h. The current change rate on the first day was 100.00%; the current change rate on the second day was 89.94%; and the current change rate on the third day was 72.46%.The above shows that the ginseng root-meristem sensor can be used stably for at least 2 days.

### 3.6. Discussion

As shown in Table 1 and Figure 4, the four-year-old ginseng root meristems had the highest sensing sensitivity to sodium nitrate, while the two- and six-year-old ginseng root meristems had a relatively low sensitivity to sodium nitrate. This trend is apparently related to photosynthesis, ginseng growth, and biomass, particularly the accumulation of active ingredients during main root growth. Inorganic nitrogen (nitrate nitrogen and ammonium nitrogen) in plants can be converted into amino nitrogen through a series of reduction reactions [33]. The amino nitrogen then stores the nitrogen source nutrients in amides through primary synthesis and forms amino acids through secondary synthesis, and consequently, plant protein biomass is synthesized. In this study, sodium nitrate was first sensed, absorbed, and metabolized by the ginseng. It then coordinated with the amino acids and nucleotide carbon skeletons formed by photosynthesis, the Calvin cycle, pentose phosphate pathway, and tricarboxylic acid cycle to autotrophically synthesize the basic raw materials (such as DNA, RNA, and protein) required for nitrogen-containing macromolecules, leading to biomass accumulation. Therefore, with increasing ginseng age in the first four years, the biomass of the main root of the ginseng increased, and the increase in biomass was accompanied by an increase in the sensitivity of inorganic nitrogen sources, such as nitrate. This process is consistent with the characteristics of ginseng whereby a large number of cells is required to multiply and increase biosynthesis activity during the fourth year of growth. However, after four years, the growth of the ginseng and the capacity for the autotrophic synthesis of amino acids and nucleotides gradually declined, and the sensitivity of the ginseng roots to inorganic nitrogen sources, such as nitrates, also gradually decreased.

The sensing trends of urea and sodium nitrate were basically similar. The sensing sensitivity of ginseng to urea showed an increasing trend (urea: Ka = 3.333 × 10^−22^ mol/L; sodium nitrate: Ka = 6.529 × 10^−24^ mol/L) in the first four years and then showed a decreasing trend after four years, and the sensing sensitivity of six-year-old ginseng was relatively low (9.970 × 10^−21^ mol/L). The above results show that through inorganic nitrogen, ginseng exhibits very strong growth vitality in transamination, deamination, ornithine cycle, and arginine and nucleotide synthesis in the first four years. These biological activities decline after four years, and senescence gradually begins. As a result, the sensing sensitivity of the ginseng root meristem to rhizosphere inorganic nitrogen also begins to decrease.

The root meristems exhibited similar sensing trends for two nucleotides, namely disodium inosinate and disodium guanylate. The two-year-old ginseng showed a relatively high sensitivity, and the sensitivity in the older ginseng decreased, but the overall trend was basically stable, suggesting that the demand for nucleotide organic nitrogen was relatively stable throughout the growth process of ginseng. Compared with those of the other nitrogen sources, the nucleotide sensing sensitivity was considerably lower, indicating that rhizosphere nucleotides are not a rigid requirement for main root growth during ginseng growth, as nucleotides can be used repeatedly as the basic materials of DNA and RNA, or alternatively, that a salvage route supply exists for nucleotide synthesis.

Sodium glutamate was the only nitrogen source nutrient whose sensing sensitivity continued to increase with increasing ginseng age, demonstrating that with increasing age, the amino acids in the rhizosphere of the ginseng gradually decreased, and the continued survival and growth of ginseng became increasingly dependent on the sensing sensitivity of the root to rhizosphere sodium glutamate. Interestingly, the magnitude of increase in sensing sensitivity was from 9.277 × 10^−19^ mol/L to 5.451 × 10^−24^ mol/L within six years; an increase of nearly five orders of magnitude.

In our previous research, we successfully constructed sex attractant hormone (bombykol) antenna and receptor sensors of male *Antheraeapernyi*, and umami, sweet, bitter, fat aroma taste tissue or receptor sensors of higher animals [2,4,17,34]. These receptors belong to the G-protein-coupled receptor superfamily. Numerous studies have proved that the sensory receptors for umami taste, sweetness, and fat aroma not only exist in the taste buds, but are also widely present in the intestine and various tissues, organs, sperm, and egg cells, and it has also been demonstrated that the umami taste receptors essentially transmit the nitrogen nutrient signal [35]. Therefore, scientists have become increasingly aware that G-protein-coupled-receptors are widely present in various cells, tissues, and organs of animals, plants, and microorganisms, and play an important role in various nutrient sensing activities [36]. Via systematic research, the authors of the current study found that the amino acid sequences of human umami receptor T1R1 and A. thaliana GCR1 share high homology. The study of the root meristem and receptor nano-gold sensor has proved that GCR1 and T1R1 are similar and that they are human umami (sodium glutamate, disodium inosinate, and disodium guanylate) sensor receptors (unpublished material). GCR1, as a type of GPCR, was first discovered in the model plant A. thaliana but has since been widely found in many plants [37,38,39]. It plays an important role in controlling physiological activities, including plant structure, nitrogen, water use efficiency, seed size and quantity, and even nitrogen fixation, as well as agronomic characteristics [9,40,41].

Based on the above analysis, we proposed the following hypothesis: the ginseng root meristem senses sodium glutamate through the GCR1 receptor, and there are two alleles in the ginseng population: R^h^ is the code for the sodium glutamate high-sensitivity receptor; R^l^is the code for the sodium glutamate low-sensitivity receptor. Therefore, there are three genotypes in the population: the R^l^R^l^low-sensitivity homozygous receptor genotype, the R^h^R^l^ medium-sensitivity heterozygous receptor genotype, and the R^h^R^h^ high-sensitivity homozygous receptor genotype. Our further hypothesis is as follow: R^h^ is not completely dominant relative to R^l^, and the phenotypes of the three genotypes are in the order of R^h^R^h^ sensing sensitivity >R^h^R^l^ sensing sensitivity >R^l^R^l^ sensing sensitivity. Due to aging of the ginseng and the sensing and absorption of sodium glutamate by the ginseng, the concentration of sodium glutamate in the rhizosphere of the ginseng continues to decrease until it drops below the sensing sensitivity (no longer capable of sensing). At this time, ginseng mortality occurs due to the sodium glutamate sensing defects. Therefore, the order of mortality is R^l^R^l^ > R^h^R^l^ > R^h^R^h^. It should be noted that the order of mortality is only statistically significant because the soil environment of the individual root of different ginseng seedlings is different, and the concentration and consumption rate of sodium glutamate are also different. With the continuous consumption of sodium glutamate, the change in its concentration also differs.

Based on the above analysis, we proposed the following hypothesis: the ginseng root meristem senses sodium glutamate through the GCR1 receptor, and there are two alleles in the ginseng population: R^h^ is the code for the sodium glutamate high-sensitivity receptor; R^l^is the code for the sodium glutamate low-sensitivity receptor. Therefore, there are three genotypes in the population: the R^l^R^l^low-sensitivity homozygous receptor genotype, the R^h^R^l^ medium-sensitivity heterozygous receptor genotype, and the R^h^R^h^ high-sensitivity homozygous receptor genotype. Our further hypothesis is as follow: R^h^ is not completely dominant relative to R^l^, and the phenotypes of the three genotypes are in the order of R^h^R^h^ sensing sensitivity >R^h^R^l^ sensing sensitivity >R^l^R^l^ sensing sensitivity. Due to aging of the ginseng and the sensing and absorption of sodium glutamate by the ginseng, the concentration of sodium glutamate in the rhizosphere of the ginseng continues to decrease until it drops below the sensing sensitivity (no longer capable of sensing). At this time, ginseng mortality occurs due to the sodium glutamate sensing defects. Therefore, the order of mortality is R^l^R^l^ > R^h^R^l^ > R^h^R^h^. It should be noted that the order of mortality is only statistically significant because the soil environment of the individual root of different ginseng seedlings is different, and the concentration and consumption rate of sodium glutamate are also different. With the continuous consumption of sodium glutamate, the change in its concentration also differs.

In order to illustrate and test this hypothesis, we collected and reviewed the existing relevant literature. Based on experimental data, starting from sowing, the number of ginseng seedlings emerging in the first year was used as the cardinal number, straight planting and no changes in the soil properties were used as the preconditions, and the seedling survival rates of five age groups were collected (Table 3). Data fitting was performed with age on the *x*-axis and mortality on the *y*-axis, as shown in Figure 5. According to the Hardy–Weinberg equilibrium, the highly sensitive receptor gene frequency of the ginseng seedling population was p, the low sensitive receptor gene frequency was q, and the genotype frequency of the seedling population conformed to the binomial theorem:(p + q)^2^ = p^2^(R^h^R^h^) + 2pg (R^h^R^l^) + q^2^(R^l^R^l^)(5)
If p ≅ q(6)

Then
p^2^(R^h^R^h^):2pg (R^h^R^l^):q^2^(R^l^R^l^) ≅ 1:2:1(7)

Based on the experimental results reported in the literature and the seedling mortality rate, Figure 5 could be obtained. At the same time, the calculation showed the ginseng mortality time periods for the R^l^R^l^ and R^h^R^l^ genotypes: the average mortality period of the R^l^R^l^ genotype population was 1 to 3.8 years (from Year 1 to Year 4), and the average mortality period of the R^h^R^l^ genotypes was 3.8 to 5.4 years (Year 4 to Year 6). The mortality rate was consistent with the predicated value.

According to the Hardy–Weinberg equilibrium p^2^(R^h^R^h^)/2pg (R^h^R^l^)/q^2^(R^l^R^l^) ≅ 1:2:1, the R^h^R^l^ genotype mortality period was roughly determined to be from Year 1 to Year 4, the R^h^R^l^ genotype mortality period was from Year 4 to Year 6, and the actual mortality rate was consistent with the predicted value.

Apparently, these experimental data were completely consistent with the data obtained by the ginseng root-meristem sensor. This preliminarily confirmed that the survival and growth of ginseng depend on the sensing sensitivity of the receptors on the root meristem to the rhizosphere sodium glutamate, the concentration of rhizosphere sodium glutamate, and the sensing and absorption of sodium glutamate, which may be the main factors causing continuous cropping obstacles in ginseng. Of course, some questions remain: Does the root system really sense sodium glutamate through GCR1? Are there two types of receptors: high sensitivity and low sensitivity? What is its mechanism of action? Why can ginseng not achieve the complete autotrophic synthesis of sodium glutamate, but instead depend on sensing and absorption from the root system? Further research is needed to answer these questions.

## 4. Conclusions

In this study, the ginseng root meristem sensor was successfully developed, and the sensing kinetics of immobilized ginseng root meristems to five important nitrogen source nutrients were detected, and the interconnected allosteric constant Ka and other important parameters were calculated. According to the sensing parameters of inorganic nitrogen sources, sodium nitrate and urea, the sensing sensitivities of two-year-old, four-year-old, and six-year-old ginseng all increased first and then decreased. The result indicated that inorganic nitrogen sources, as the basic raw material of protein and nucleic acids, are essential at the stage when ginseng biomass is increasing and they coordinate with the carbon skeleton, which is synthesized and transformed by photosynthesis. While the sensing sensitivities to disodium guanylate and disodium inosinate were relatively low, they did not change considerably with increasing age. The above finding suggests that nucleotide organic nitrogen, as the basic raw material of genetic material, can be reused through salvage routes during the survival and growth of ginseng, and basically does not depend on the sensing and absorption of nucleotide organic nitrogen by roots. It is very significant that the sensitivity of the immobilized root meristem to sodium glutamate (1/Ka) increased by nearly five orders of magnitude with increasing ginseng age. The above result indicates that the survival and growth of ginseng largely depend on the roots to sense and absorb sodium glutamate in the rhizosphere. In summary, the ginseng root meristem sensor provides a highly sensitive platform for the study of the ability of ginseng rhizosphere to sense nitrogen source nutrition. Our analysis of the existing research and the mortality rate data reported in the literature during the cultivation process of ginseng preliminarily proved that the sensitivity of the sensing and absorption of sodium glutamate in the rhizosphere may be an important factor in the continuous cropping obstacles of ginseng.

## Figures and Tables

**Figure 1 sensors-21-00681-f001:**
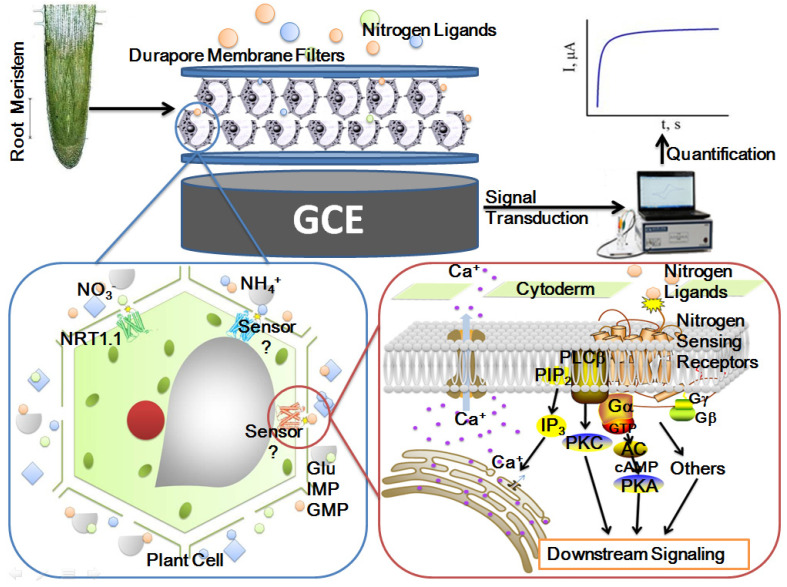
Diagram of the measurement principle of the ginseng root-meristem sensor.

**Figure 2 sensors-21-00681-f002:**
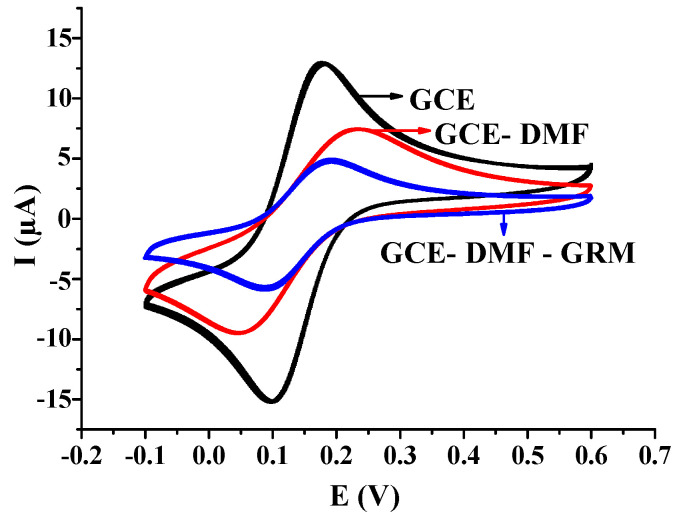
Cyclic voltammetry characterization of the ginseng root-meristem sensor. Abbreviation: GCE, glassy carbon electrode; DMF, durapore membrane filters; GRM, root meristem of Panax ginseng.

**Figure 3 sensors-21-00681-f003:**
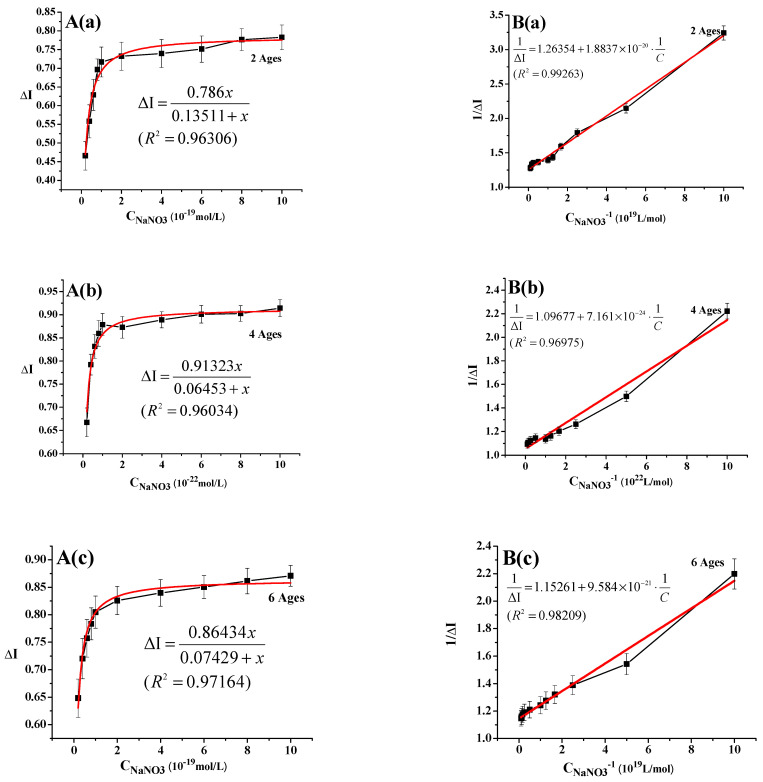
Linear relationship between the response current and the concentration of sodium nitrate: (**A**) Determination of the correlation curve between the response current of the root meristem sensors of two-, four-, and six-year-old ginseng plants and the concentration of sodium nitrate; (**B**) Double reciprocal regression curves of the root-meristem sensors of two-, four-, and six-year-old ginseng plants in sensing a sodium nitrate solution.

**Figure 4 sensors-21-00681-f004:**
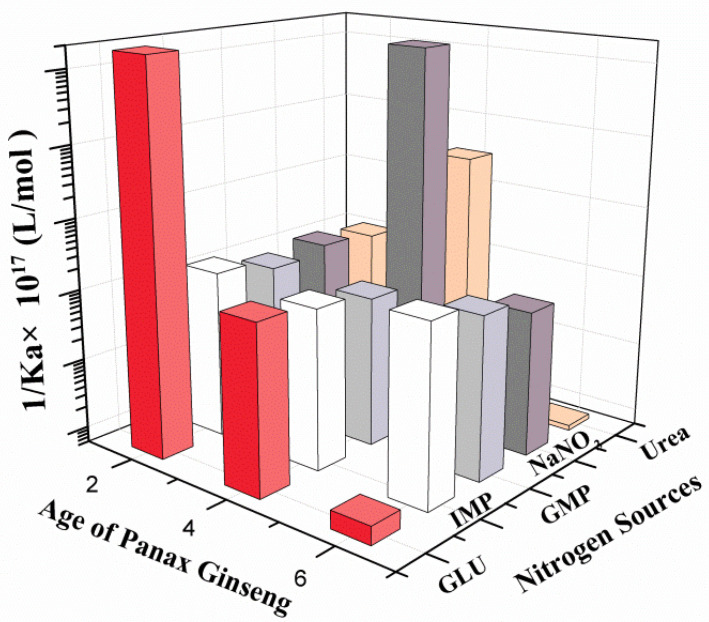
Column chart comparing the sensitivities (1/Ka) of the ginseng root meristems at each age to five different nitrogen sources.

**Figure 5 sensors-21-00681-f005:**
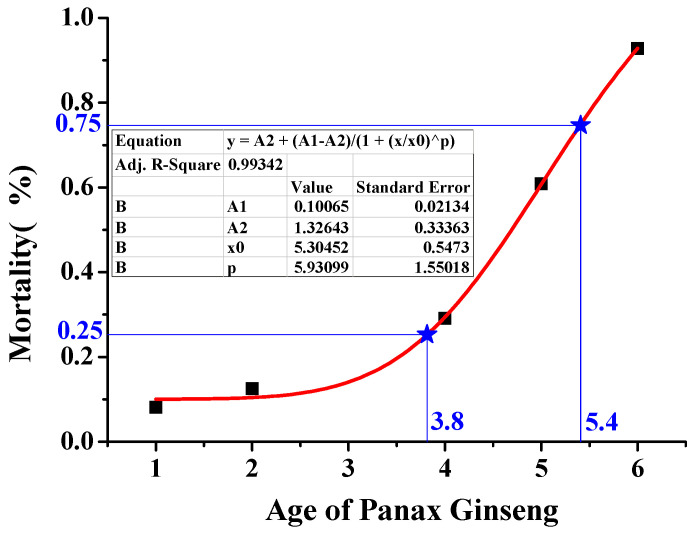
Mortality period of ginseng with R^l^R^l^ and R^h^R^l^ genotypes. Ginseng age is on the *x*-axis and mortality rate is on the *y*-axis. According to the Hardy–Weinberg equilibrium p2(R^h^R^h^)/2pg (R^h^R^l^)/q2(R^l^R^l^) ≅ 1:2:1, the mortality period of the R^l^R^l^ genotype was roughly determined to be Year 1 to Year 4, the mortality period of the R^h^R^l^ genotype was from Year 4 to Year 6, and the actual mortality rate was consistent with the predicted value.

**Table 1 sensors-21-00681-t001:** The interconnected allosteric constant Ka values of the root meristems of three types of ginseng plants in sensing five types of nitrogen source nutrients.

Age of Ginseng (Years)	Ka × 10^−21^ (mol/L)				
	IMP	GMP	Sodium Glutamate	NaNO_3_	Urea
Two	5.024	8.339	927.7	14.91	1416
Four	9.035	12.44	6.980	0.006529	0.3333
Six	7.035	10.67	0.005451	8.315	9.970

**Table 2 sensors-21-00681-t002:** Comparison of various detection methods available.

Method	Sample	Detection Target	Detection Limit (mol/L)	Reference
Catalytic-spectrophotometric	Water/Food	Nitrate	4.838 × 10^−4^	[28]
Atomic absorption spectroscopic	Water	Nitrate	8.063 × 10^−4^	[29]
Fluorescence-spectroscopic	Water/Soil/Forensic samples	Nitrate	1.0 × 10^−5^	[30]
Chemiluminescence	Water	Nitrate	3.2 × 10^−7^	[31]
Electrochemical conducting polypyrrole films	Nitrate	Nitrate	4.7 × 10^−5^	[32]
Electrochemicalroot-meristem sensor	Ginseng Root-Meristem	Pure solutions of various nutrients	6.529 × 10^−24^	This work

**Table 3 sensors-21-00681-t003:** Data investigation on the seedling survival rate of ginseng plants of five ages.

Age of Ginseng (Years)	Cardinality (Emergence of Seeding Rate)	Survival Number	Survival Rate	Mean	Mortality	References
One	1128	1004	89.01%	91.89%	8.11%	[10]
	1128	1032	91.49%			
	1142	1087	95.18%			
Two	1128	977	86.61%	87.45%	12.55%	
	1128	963	85.37%			
	1142	1032	90.37%			
Four	16.8 (AVG)	11.9 (AVG)	70.83%	70.83%	29.17%	[42]
Five	935	330	35.29%	39.13%	60.87%	[10]
	922	344	37.31%			
	922	413	44.79%			
Six	935	49	5.24%	7.24%	92.76%	
	922	55	5.97%			
	922	97	10.52%			

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
