# Peer review of "Construction of a Ginseng Root-Meristem Sensor and a Sensing Kinetics Study on the Main Nitrogen Nutrients"

_sensors, 2021, doi:10.3390/s21030681_

Round 1

Reviewer 1 Report

The article is very interesting and can be published as it

Author Response

Thanks reviewer so much!

Reviewer 2 Report

December 31th, 2020.

Mr. Cigar Zeng
Assistant Editor

Sensors Editorial Office

Correspondence reference: sensors-1051939

“Construction of a Ginseng Root-Meristem Sensor and a Sensing Kinetics Study on the Main Nitrogen Nutrients”

Dear Editor,

The manuscript, “Construction of a Ginseng Root-Meristem Sensor and a Sensing Kinetics Study on the Main Nitrogen Nutrients” describes a very confuse ginseng root-meristem sensor. The results are not interesting. The paper is unclear and the experimental procedures are not described in sufficient detail. Any picture of systems was showed in the paper. The reproducibility studies, control experiments and long-term stability were not made. The manuscript is not suitable for the scope of the Sensors. "Results and Discussion" needs to be more complete with some investigations. I recommend rejection this manuscript version with special attention for:

  1. Lines 15 and 21-22: limit of detection values did not show units.
  2. Line 112: “−0.1–0.6 V”: -0.1 V until to -0.6 V or -0.1 V until to +0.6 V?
  3. Lines 112-116: Ferrocyanides redox probes are strongly adsorbed on glassy carbon electrodes or any other carbon surfaces. Electrodes washed with ultrapure water is not sufficient to desorption of redox probes.
  4. Line 118: What was the source of starch?
  5. In Figure S6, there are several linear fitting using points non-linear.
  6. The manuscript is confusing.

Author Response

AThanks for your comments. The aim of this study was to detect the sensing ability of ginseng root meristem to different types of nitrogen nutrients by ginseng root meristem sensor. And then, it is scientifically inferred that the decrease of ginseng's ability of sensing and synthesizing glutamate has an important effect on the continuous cropping of ginseng. Therefore, this paper discusses the importance of plants sensing nitrogen nutrition and the basis for the selection of these five nitrogen nutrients (see Lines 29-78, Section 1),and describes the experimental materials and processes one by one according to the experimental procedures (see Lines 80-173, Section 2). In addition, as for the Repetitive study of the experiment, the "Repetitive Runs" parameter is set to "3 Number of Runs" when the CHL600E electrochemical workstation is used to record the data in this study, therefore, the data in this paper have been repeated for 3 times (see Line 154, Section 2.5); as for the control experiment, this article Lines 147-158 has been pointed out that the response current I1 obtained by detecting ultrapure water was taken as the blank control for each data; as for the reproducibility and stability of this study, we have added in the paper and marked the content in red(Lines 294-305).

  • Lines 15 and 21-22: limit of detection values did not show units.

A: Thanks reviewer’s good suggestion. We have added the unit after limit of detection values. And all units in the Manuscript have been modified carefully.

  • Line 112: “−0.1–0.6 V”: -0.1 V until to -0.6 V or -0.1 V until to +0.6 V?

A: We accepted. "-0.1–0.6 V" means -0.1 V until to +0.6 V, and this has been modified in red, too.

3Lines 112-116: Ferrocyanides redox probes are strongly adsorbed on glassy carbon electrodes or any other carbon surfaces. Electrodes washed with ultrapure water is not sufficient to desorption of redox probes.

A: Thanks reviewer’s good suggestion. We have modified this description. According to the report in the reference, we first pretreated several electrodes at the same time, and then took out one of these electrodes and placed it in a 1×10-3mol/L K3Fe(CN)6 solution containing 0.20 mol/L KNO3. Characterization to prove that this pre-processing operation is feasible;the other electrodes were not placed in the K3Fe(CN)6 solution for characterization, but directly assembled the sensor. All revised words were shown in red color (Lines 103-117).

4Line 118: What was the source of starch?

A: Thanks reviewer. The Soluble starch was provided from Yingda Sparseness & Nobel Reagent Chemical Factory (Tianjin, China) (Lines 85-86).

5In Figure S6, there are several linear fitting using points non-linear.

A: Thanks reviewer’s suggestion. Figure S6 is based on the average of the original data. Use the originPro 8 software to draw the graph after double reciprocal value. Then use linear fitting to process the data. After linear fitting, all R2 > 87%. Therefore, overall, the trend of the points are still in linear. In addition, although there are random effects in the experiment, according to a large amount of work published in the past (such as References[2,4,5-7]) and comparing the kinetic equation of the enzyme, we believe that the parameters obtained by linear fitting the data by the double reciprocal method are reasonable. For this part of the content, we have expressed red in the article(Lines 255-256, Line 270).

6The manuscript is confusing.

A: Thanks reviewer. We have trimed the units, expressions and structure of the article. And we very much hope to meet the requirements of the magazine.

Reviewer 3 Report

The manuscript by Shiang Wang et al can be characterized as a typical contribution to the wide database of similar articles concerning the development, on site disposable testing, and model applications of electrochemical biosensors intended for nitrogen determinations. Moreover, this article is another contribution dealing for five important nitrogen nutrients by ginseng plants. With respect to these features, the paper seems to be potentially attractive to the readers of Sensors, reporting widely on research activities of such a kind. Regarding the manuscript itself, it is written quite well, but it comprises one very problematic section concerning detection concentration throughout the paper, which may not express clearly to the concentration levels of nitrogen. Furthermore, except ABSTRACT shows the sensitivity of the sensor, which lack of the unit and the detection target. Selection of graphic and tables seems to be okay and also the list of references has been assembled appropriately. By summing up, the paper by Shiang Wang et al can be recommended for publication after major revision and conditionally, if the revised work will contain newly performed and commented determination experiments and interference studies. Otherwise, the overall revision should also reflect the following points: 1) Principal comment: Firstly, all of the numbers, illustrating sensitivity and detection limit, should have appropriate units and all the target measurements should calculated by the same nitrogen element since organic nitrogen nutrients are not in the form of sodium nitrate in natural. Although some of the references mentioned here are significant, other representative paper(s); best recent methods should also be cited here as the summary of nitrate detection. 2) Graphics; Before Fig. 2, how was the SEM of the “sandwich” structure of the ginseng root-meristem sensor? Moreover, the voltage peak-to-peak value (∆Ep) of the CV characterization for GCE, GCE-DMF, GCE-DMF-GRM electrode might highly indicate the influence of different materials for electric double layer on the surface of sensor. 3) Fig 3 A(b) and A(b) are the same figure. Lacking of Double reciprocal regression curves of the root-meristem sensors of four-year-old ginseng plants. Logarithmic expressions between current and sodium nitrate concentration are commonly used for calibration equations in section 3.4. 4) Results and Discussion Table 1;. As the authors are well aware, the detection measurements are various; and the number of different possibilities for the ginseng root-meristem sensor could reach a lower limit of detection (LOD), especially for sodium nitrate detection. Furthermore, there have already been hundreds of studies reporting sodium nitrate sensor which could determinate trace level accurately comparing AAS. How was the results among the AAS, electrochemical methods and this work for nitrate determination? 5) Results and Discussion; Is there cross-interferences from natural environment, such as phosphate, sodium, chloride or any other ions? And how to reduce the interference for nitrate determination?

Author Response

AThanks reviewer’s positive comments and suggestions. We have modified this description. Below you will find our point-by-point responses to their comments.

1) Principal comment: Firstly, [a]all of the numbers, illustrating sensitivity and detection limit, should have appropriate units and [b]all the target measurements should calculated by the same nitrogen element since organic nitrogen nutrients are not in the form of sodium nitrate in natural. Although some of the references mentioned here are significant, other representative paper(s); best recent methods should also be cited here as the summary of nitrate detection.

A: Thanks reviewer’s good suggestion.

[a] We have added the unit after sensitivity and detection limit. And all units in the Manuscript have been modified carefully.

[b] We also considered this question when designing the experiment. Accordingly, we paid particular attention when the interaction between the sensing receptors and the ligands, is the ligands used for signal transmission in the form of nitrogen element or molecular? Answer: For inorganic nitrogen, the ligands tap on the sensing receptor in the form of ions to transmit signals, such as NO3- and NH4+; for organic nitrogen, the ligands are in the form of molecules. Therefore, in accordance with past experience, mol/L is used as the unit of calculation continually, instead of the nitrogen content. We have added the relevant discussion and expressed in red (Lines 224-227).

2) Graphics; [a]Before Fig. 2, how was the SEM of the “sandwich” structure of the ginseng root-meristem sensor? [b]Moreover, the voltage peak-to-peak value (∆Ep) of the CV characterization for GCE, GCE-DMF, GCE-DMF-GRM electrode might highly indicate the influence of different materials for electric double layer on the surface of sensor.

A: Thanks reviewer’s good suggestion.

[a] The ginseng root-meristem sensor is a kind of biological tissue sensor, which does not reach the molecular level. Hence scanning electron microscopy is not necessary.

[b] Indeed, the combination of different materials on the electrode has an effect on ∆Ep, but it has no effect on the actual detection.

3) Fig 3 A(b) and A(b) are the same figure. Lacking of Double reciprocal regression curves of the root-meristem sensors of four-year-old ginseng plants. Logarithmic expressions between current and sodium nitrate concentration are commonly used for calibration equations in section 3.4.

A: Thanks reviewer’s good suggestion. This picture was uploaded incorrectly. We have amended the figure B(b) in the manuscript (Line 240).

4) Results and Discussion Table 1;. As the authors are well aware, the detection measurements are various; and the number of different possibilities for the ginseng root-meristem sensor could reach a lower limit of detection (LOD), especially for sodium nitrate detection. Furthermore, there have already been hundreds of studies reporting sodium nitrate sensor which could determinate trace level accurately comparing AAS. How was the results among the AAS, electrochemical methods and this work for nitrate determination?

A: Thanks reviewer’s good suggestion. As one of many sensors, the ginseng root-meristem sensor should indeed be compared with other detection methods. We have added a part of the discussion and expressed it in red (Lines 271-276). But we also need to emphasize that other detection methods are used to detect the content of sodium nitrate in the sample. The ginseng root-meristem sensor is used to detect the sensing ability of plant roots to nutrients, such as sodium nitrate. So the targets and intentions of the detections are different.

5) Results and Discussion; Is there cross-interferences from natural environment, such as phosphate, sodium, chloride or any other ions? And how to reduce the interference for nitrate determination?

A: Thanks reviewer. The ginseng root-meristem sensor does have certain interference, such as ionic strength, pH value, etc. In order to avoid such interference, all the water used in the experiment is ultra-pure water, including the solvent of the prepared solution, blank control water, soaking and cleaning water; and the measurement target of the experiment is as pure as possible, so as to minimize the interference of the experiment.

Round 2

Reviewer 2 Report

January 17th, 2021.

Sensors Editorial Office

Correspondence reference: sensors-1051939-peer-review-v2

Construction of a Ginseng Root-Meristem Sensor and a Sensing Kinetics Study on the Main Nitrogen Nutrients

Dear Editor,

The manuscript “Construction of a Ginseng Root-Meristem Sensor and a Sensing Kinetics Study on the Main Nitrogen Nutrients” was meticulously corrected and improved by the authors. Now, the manuscript can be accept for publication is its current form.
